# Maternal HPV Infection and the Estimated Risks for Adverse Pregnancy Outcomes—A Systematic Review

**DOI:** 10.3390/diagnostics12061471

**Published:** 2022-06-15

**Authors:** Simona Daniela Popescu, Andreea Gratiana Boiangiu, Romina-Marina Sima, Liviu Bilteanu, Simona Vladareanu, Radu Vladareanu

**Affiliations:** 1Neonatology Clinic, Department of Obstetrics and Gynecology, Faculty of General Medicine, Elias University Hospital, Carol Davila University of Medicine and Pharmacy, 011461 Bucharest, Romania; simona.popescu@umfcd.ro; 2Obstetrics and Gynecology Clinic, Department of Obstetrics and Gynecology, Faculty of General Medicine, Elias University Hospital, Carol Davila University of Medicine and Pharmacy, 011461 Bucharest, Romania; radu.vladareanu@umfcd.ro; 3Obstetrics and Gynecology Clinic “Bucur” Maternity, Department of Obstetrics and Gynecology, Faculty of General Medicine, Saint John Hospital, Carol Davila University of Medicine and Pharmacy, 011461 Bucharest, Romania; romina.sima@umfcd.ro; 4Department of Preclinic Sciences, Faculty of Veterinary Medicine, University of Agronomic Sciences and Veterinary Medicine of Bucharest, 050097 Bucharest, Romania; liviu.bilteanu@fmvb.usamv.ro; 5Laboratory of Molecular Nanotechnologies, National Institute for Research and Development in Microtechnologies, 077190 Bucharest, Romania

**Keywords:** HPV, miscarriage, preterm birth, premature rupture of membranes, preeclampsia, fetal growth restriction

## Abstract

Background: Human Papilloma Virus (HPV) represents the most prevalent genital infection in young women of reproductive age. Objective: This systematic review aims to estimate the effect of HPV infection during pregnancy and assess the correlation between HPV and adverse pregnancy outcomes. Materials and methods: The search strategy has been developed based on the PICOS framework: Population (pregnant women infected with HPV), Intervention (HPV infection confirmed by molecular tests), Comparator (pregnant women without HPV infection), Outcomes (adverse pregnancy outcomes) and Study design (observational studies). We searched PubMed, Web of Science, and Scopus databases on 8 January 2022 by using the following keywords: “HPV”, “prematurity”, “preterm birth”, “miscarriage”, “premature rupture of membranes”, “adverse pregnancy outcome”, “low birth weight”, “fetal growth restriction”, “pregnancy-induced hypertensive disorders”, “preeclampsia”. Selection criteria were HPV infection confirmed within maximum 2 years before pregnancy with a molecular test and adverse pregnancy outcomes. (Results: Although numerous studies are conducted on this topic, data are still controversial regarding identifying maternal HPV infection as a risk factor for adverse pregnancy outcomes. More prospective large cohort studies are needed to prove a causative relationship.

## 1. Introduction

HPV is a double-stranded DNA virus from the Papillomaviridae family, specific to humankind. It is composed of a family of over 200 subtypes with specific tissue tropism (either cutaneous or mucosal) [1]. A total of 40 subtypes are recognized to cause genital infections. HPV is divided into 2 categories based on the oncogenic potential: high risk category (HR-HR-HPV) causing premalignant lesions or cancers of the anogenital area (16, 18, 31, 33, 35, 39, 45, 51, 52, 56, 58, 59, 68, 73, and 82) and low risk category (LR-LR-HPV) causing papilloma’s and genital warts (6, 11, 40, 42, 43, 44, 54, 61, 62, 71, 72, 81, 83, 84, 85, 89) [1]. Worldwide, HPV infection is one of the most prevalent infections in both women and men of fertile age, according to the CDC [2,3]. The link between HR-HR-HPV infection and cancers is well established regarding carcinoma of the anogenital area: i.e., cervical, vulvar and colorectal and of the head and neck area, i.e., oropharyngeal carcinomas [4,5,6,7,8] and penile cancer [9,10].

Pregnant women, due to their immunological and endocrinological status, have a higher risk of HPV infection, with an infection rate reported to be 82% compared with a rate of 10.4% in non-pregnant women [11]. HPV cervical infection has a higher rate of detection during pregnancy, i.e., 15.53% compared to 12.6% in non-pregnant population [12,13]. Usually, HPV infection is a transient and auto-limitative, but may be persistent in immunosuppressed patients, and pregnancy should be rendered as an immunosuppressive state. Regarding the vertical transmission of the HPV infection from the mother to the newborn, there is important evidence that cesarean section has a protective role in the infection with a significantly lower transmission risk [14,15]. Intrauterine HPV infection may also occur as HPV DNA was identified in the placenta, the amniotic fluid, the membranes [16,17]. Intrauterine fetal infection can occur also by ascending infection thru the vagina and cervix, or by hematologic spread [18,19]. There is evidence that proves that HPV can infect the trophoblastic cells [20,21,22]; therefore the pregnancy may be affected the via the intrauterine environment. The correlation between HPV infection and poor pregnancy outcomes such as spontaneous miscarriage, preterm birth, preeclampsia, preterm premature rupture of membranes, intrauterine growth restriction, low birth weight is a topic of high interest [23,24,25] Starting from the theory that any disruption of the placental environment can be caused by diverse infection and had the potential to induce adverse pregnancy outcomes and the findings from in vitro studies that have demonstrated that HPV may infect the trophoblastic cells, results that HPV infection during pregnancy, as a viral infection, may be associated with adverse pregnancy course [12,23,25,26].

Viral infections are recognized as a risk factor for adverse pregnancy outcomes [27], and as HPV infection is the most prevalent genital viral infection among females of reproductive age, the association between HPV infection and adverse pregnancy outcomes should be carefully investigated, and this is the aim of our systematic review.

## 2. Materials and Methods

This systematic review is conducted following the Preferred Reporting Items for Systematic Reviews and Meta-Analyses guidelines [28] and aims to estimate the effect of HPV infection during pregnancy and assess the correlation between HPV and adverse pregnancy outcomes.

### 2.1. Search Strategy

On 8 January 2022, we searched several databases: PubMed, Web of Science and Scopus, targeting literature published between 1 January 2012 and 1 January 2022. We used the following keywords: “HPV”, “prematurity”, “preterm birth”, “miscarriage”, “premature rupture of membranes”, “adverse pregnancy outcome”, “low birth weight”, “fetal growth restriction”, “pregnancy-induced hypertensive disorders”, “preeclampsia”.

### 2.2. Eligibility

The search strategy has been developed based on the PICOS framework: Population (pregnant women infected with HPV), Intervention (HPV infection confirmed by molecular tests), Comparator (pregnant women without HPV infection), Outcomes (adverse pregnancy outcomes), and Study design (observational studies). The inclusion criteria applied for the elected population were: HPV infection confirmed within maximum two years before pregnancy or during pregnancy with a molecular test and adverse pregnancy outcomes.

### 2.3. Exposure

The exposure of interest is the association of HPV infection in a pregnant women with adverse pregnancy outcome. The presence of HPV infection in all genital sites (vulvar, vaginal, or cervical) must be confirmed by a molecular test within maximum two years before pregnancy or during the pregnancy.

The comparison group will include pregnant women that are not HPV infected or are not documented with infection.

We have excluded: editorials, opinions, case reports, studies without a control group, and studies in which HPV infection was not molecularly documented. We have also excluded studies that explored the post-surgical treatment of cervical HPV during pregnancy associated with adverse pregnancy outcomes, post-vaccine studies, HPV sperm-related studies, and co-infection studies.

The primary outcome of this review is the link between HPV infection and premature birth. The secondary outcomes were miscarriage, premature rupture of membranes (PROM), pregnancy-induced hypertensive disorders (PIHD), and fetal growth restriction (FGR).

Two reviewers (SDP and AGB) independently screened titles/abstracts independently and in duplicate and extracted data from selected full-text reports using a predesigned form. Disagreements were resolved by discussion with the coauthors.

We included in this systematic review 21 studies. From which 3 were systematic reviews and meta-analyses [13,29,30], 4 reviews articles [31,32,33,34], 5 were retrospective cohort studies [35,36,37,38,39], 2 prospective cohort studies [23,24], 4 case-control studies [12,31,40,41], 2 cross-sectional studies [42,43] and 1 data-linkage study [44] Figure 1.

## 3. Results

### 3.1. Primary Outcome—Preterm Birth

Preterm birth is assumed as any birth before 37 weeks and after 24 weeks of gestation. About 15 million newborns are born preterm worldwide, with a higher incidence in low- and middle-income countries (LMIC). It is the primary cause of an estimated one million neonatal deaths annually, and is a significant contributor to childhood morbidity. Starting from the fact that intrauterine infections act as a determinant factor for adverse pregnancy outcomes, especially for preterm birth, numerous studies have tried to investigate the relationship between maternal HPV infection and adverse pregnancy outcomes. However, not all studies published on this theme have proven this correlation keeping controversies in this field.

A number of 11 studies have investigated as primary outcome the link between preterm birth and HPV infection [13,24,29,30,31,32,33,38,40,44,45].

You et al. [22] have demonstrated that when in vitro trophoblastic cells are cultivated with HPV subtypes 16, 18, 11 and 31 there is an active viral expression on both early and late genes that reduce the number of trophoblastic cells and weakens the trophoblast-endometrial cell adhesion. In terms of genomic structure, HPV is a double-stranded DNA virus divided into three parts: the early gene region (E) with a role in viral transcription, the late gene region (L) with a role in generating the capsid proteins, and the upstream regulatory region (URR) without a role in protein coding, but containing about 10% of the viral genome and with a keratinocyte-specific enhancer region [46]. Those in vitro findings were similar to other studies [47] that demonstrated an increase in the apoptosis rate of the infected trophoblastic cells. Trophoblastic infection may alter the embryo quality and the nesting process and, therefore directly alter the pregnancy outcomes [25,47,48]. However, the potential of placental dysfunction to induce miscarriage, preterm birth, and intrauterine growth restriction is well established, so there is not a surprise that some observational studies have identified HPV infection as a risk factor for adverse pregnancy outcomes.

Slatter et al. [43] in their cross-sectional study on 339 placentae (253 HPV positive and 86 HPV negative) have identified HPV placental infection as a risk factor for prematurity (s.d 29.2% vs. 16.3%), intrauterine growth restriction (s.d 22.4% vs. 19.8%), chorioamnionitis (s.d 9.1% vs. 2.3%) and preeclampsia (s.d 7.9% vs. 0%).

Niyibizi J et al. [30] in a lately systematic review and meta-analysis on the association between maternal HPV infection and adverse pregnancy outcomes have presented HPV infection as a risk factor for preterm birth with an age adjusted odds ratio (aOR) (aOR = 1.5; 95% CI = [1.19, 1.88]), preterm premature rupture of fetal membranes (PPROM) (aOR = 1.96; 95% CI = [1.11, 3.45]), premature rupture of membranes (PROM) (aOR = 1.42; 95% CI = [1.08–1.86]), intrauterine growth restriction (FGR) (aOR = 1.17; 95% CI = [1.01, 1.37]), low birth weight (LBW) (aOR 1.91; 95% CI 1.33–2.76) and intrauterine fetal death (IUFD) (aOR 2.23; 95% CI 1.14–4.37). In the HERITAGE cohort study Niyibizi J. et al. [26] suggested that HPV is associated with preterm birth, there is a possible association between HPV and premature rupture of membranes, however regarding miscarriage, intrauterine growth restriction and stillbirths due to the small number of studies and bias, investigations are still needed before drawing any conclusions. Also, he found no association with pregnancy-induced hypertensive disorders [30]. Further studies should be conducted using HPV test in early pregnancy in order to be able to catch the possible complications during the first trimester, i.e., early miscarriage. All future studies that will be conducted should also consider the clearance rate of HPV during pregnancy and the hormonal-dependent HPV susceptibility during pregnancy, resulting from this the importance of timing in HPV detection and the need for repeated tests during pregnancy [31].

Mosbah et al. [40] conducted an observational comparative case-control study on 103 Egyptian pregnant patients, from these 53 have preterm delivered and 50 have term delivered. The authors have identified HPV placental infection in 18.9% of women whom preterm delivered and in 4% of women who term delivered (*p* = 0.019). 60% of HPV positive women were HR-HPV (16.18) infected and 40% were LR-HPV (6.11) infected, so they finally concluded that HPV infection represents a risk factor for preterm birth among the Egyptian population, especially HR-HPV infection [40].

In another systematic review and meta-analyses that included seven studies and 45,603 patients conducted by Wu et al. [13] published in 2021, HPV infection was found to be a significative risk factor for preterm birth (OR = 1.81, 95% CI: 1.25–2.62, Z = 3.16, *p* = 0.002), and premature rupture of membranes (OR = 1.74, 95% CI: 1.45–2.10, Z = 5.84, *p* < 0.00001) but the reliability of the results may be affected due to differences in retrieval mechanisms, differences in risk factors in distinct references and potential publication bias. The authors recommend a multi-center, large sample original research in order to prove the theoretical data [13].

Into another systematic review and meta-analyses [29] the authors included 18 studies and analyzed them in order to estimate the impact of HPV on spontaneous miscarriage, preterm births and the rate of pregnancy obtained post IVF techniques and the rate of miscarriage in this subgroup. The main statistical tool calculated the OR and RR, with the 95% CI. They have not found any statistical association between HPV infection and spontaneous miscarriage with a calculated OR of 1.40, (95% CI 0.56–3.50). However, indiscriminate HPV genotype can increase the abortion rate with an OR of 2.24 (95% CI 1.37–3.65), while HR-HPV infection was found to have no significant effect (OR 0.65, 95% CI 0.21–1.98). HR-HPV infection is a risk factor for preterm births with a pooled OR of 2.84 (95% CI 1.95–4.14) [29]. Regarding pregnancy rate after IVF techniques (RR 1.04, 95% CI 0.64–1.70) and miscarriage in those patients (RR 1.47, 95% CI 0.86–2.50) HPV infection seem to have no impact and to be an independent factor [29].

Other studies have identified no link between HPV infection and adverse pregnancy outcomes i.e., preterm birth and miscarriage. Ambühl et al. [45] in a Danish cohort study have investigated the placental HPV infection and concluded that HPV placental infections are not likely to be an independent risk factor for spontaneous miscarriage or preterm birth Aldhous et al. [44] have conducted a data linkage study trying to demonstrate a relationship between maternal HPV and preterm birth and concluded that HR-HPV that induces high grade lesions of the cervix may correlate with preterm birth, but not just any HPV infection. Subramaniam et al. [38] after conducting a retrospective cohort study on 2321 patients, have also concluded that there is no link between HPV infection, and preterm birth. Some other authors in their work investigating placental HPV infection concluded that further research are needed to prove that HPV is a determinant agent for adverse pregnancy outcomes [38].

Caballero et al. [35] conducted a retrospective cohort study of singleton deliveries on a cohort of 2153 women, of which 38.5% were HPV positive. They identified HPV as a risk factor for preterm rupture of membranes, but HPV does not appear to have any additional effect on preterm delivery outside the increased rate for preterm delivery after premature rupture of the membrane [35].

### 3.2. Secondary Outcomes

The link between HPV infection and spontaneous miscarriage is investigated in 9 studies [12,23,29,30,34,37,41,43,45] the linkage with premature rupture of membranes in 6 studies [13,24,30,35,38,42], pregnancy induced hypertensive disorders are investigated in 4 studies [36,38,39,43], fetal growth restriction (FGR) in HPV positive patient is investigated in 3 studies [30,33,43] and pregnancy rate after IVF techniques and miscarriage after IVF in HPV positives are investigated in one systematic review and meta-analyses [29].

#### 3.2.1. Miscarriage

Miscarriage or spontaneous abortion is defined as a pregnancy loss before 20 weeks gestation. The World Health Organization (WHO) estimates that about 26% of all pregnancies miscarriages. Miscarriage etiology is multifactorial being involved and demonstrated genetic factors, teratogen factors, immunological factors, anatomical factors, environmental factors and infectious factors. Many studies in the field have tried to assume HPV infection as a risk factor for miscarriage, but the data up to date are controversial. Due to the immunological status of a pregnant patient and the proof that HPV may infect the cervix, the placenta, the membranes, and the amniotic fluid, the concern of researchers regarding HPV involvement in miscarriage is legitimate.

Ambühl, in the systematic review conducted, have identified a higher HPV incidence among women with miscarriage with an incidence of 24.9% HPV positive, but without assigning a clear, determining role of HPV infection [31]. Similar findings were presented by Bober et al. [12] who observed in their study a higher incidence of HR-HPV trophoblastic infection post spontaneous miscarriage compared with ordinary pregnancy course, suggesting the possible hematogenous route of infectious [12]. In addition to these studies, other researches have not identified a link between HPV infection and miscarriage. The study of Bober included 143 pregnancies, from which 84 pregnancies ended in miscarriage, and 59 had a normal pregnancy courses. They have searched HPV in the cervix and in the placenta in all 143 patient but have had identified HPV in 18% of cases in the group of miscarriage and 7% in the control group, and concluded that HPV is more prevalent among patients with abnormal pregnancy course, but the study is limited due to the small number of patient included [12]. Tognon and his coworkers have recently investigated the association between HPV infection and spontaneous abortion by testing HPV DNA presence in chorion villi and peripheral blood mononuclear cells from 80 women who have had spontaneous abortion and from other 80 women who have voluntary interrupt their pregnancy course (case controls); they have detected by using both a qualitative PCR and quantitative droplet digital PCR (ddPCR) a low rate of HPV DNA into the chorion villi and have demonstrated a low tropism of HPV virus for the chorion villi with a non-significant role in spontaneous abortion [49].

No significant association between HPV infection and miscarriage has been reported by Niyibizi in his systematic review and meta-analyses [30].

Basonidis have resumed in his work several studies regarding HPV correlation with spontaneous abortion and presented conflicting findings and limited conclusions due to the small sample size and/or methodological limitations of the studies they included. The link between HPV and miscarriage is still unclear, and more studies are needed [34].

Conde-Ferraez et al. [41] have conducted a case control-study including 281 women in Mexico to investigate if HPV cervical infection is associated with spontaneous abortion. HPV was positive in 24.4% of study cases and in 15.2% of control cases, and they found no correlation between HPV infection and spontaneous abortion

Regarding recurrent miscarriage and HPV infection, Ticconi et al. [37] found lower HPV infection rates among patients with recurrent miscarriage

The data published to date are contradictory and limited by the small sample size. More studies are needed to prove a correlation between maternal HPV infection and the risk for spontaneous abortion.

#### 3.2.2. Premature Rupture of Membranes

Premature rupture of membranes (PROM) is defined as the rupture of amniotic membranes before labor onset; it mostly occurs near term. If the membrane rupture occurs before 37 weeks of gestation, it is called preterm PROM (PPROM). Preterm PROM complicates approximately 3 percent of all pregnancies and is the generating cause of one-third of preterm births [50]. Both Preterm PPROM and near-term PROM can cause both maternal and fetal morbidity. The maternal complications that might arise from PROM or PPROM are represented by infectious complications like chorioamnionitis and placental abruption. Fetal complications that might arise from PROM or preterm PROM are: infectious complications, pulmonary hypoplasia, umbilical cord prolapse or umbilical cord compression with fetal distress and hypoxemia generating neurological impairment.

The linkage of maternal HPV infection with premature rupture of membranes is evaluated in 6 studies [13,24,30,35,38,42]. Cho et al. [42] al have investigated the incidence rates of PROM in patient with HPV in a cross sectional study that have included 311 pregnant women who gave birth at Korea University Medical Center and discovered in their cohort an increased risk of HPV infection in the PROM group—27.3% of HR-HPV-positive women experienced PROM, compared with 14.2% in the HR-HPV-negative group (*p* = 0.029) [42]. Similar findings are presented also in the studies of Cotton-Caballero’s, Pandey’s and Wiik’s [23,25,35]. In his systematic review and meta-analyses Wu analyzed 7 studies and concluded that pregnant women with HR-HPV infection have a higher risk of PROM compared with the negative control group (OR = 1.74, 95% CI: 1.45–2.10, Z = 5.84, *p* < 0.00001) [13] Niyibizi and his coworkers found a consistent and significant association between HPV infection and PROM (aOR = 1.42, 95% CI =1.08–1.86) [30] Therefore, HPV infection may represent a potential risk for PROM, and HPV testing must be taken into consideration in pregnant women or women who are willing to conceive; nevertheless genital concurrent infections must be searched.

#### 3.2.3. Pregnancy-Induced Hypertensive Disorders

Pregnancy-induced hypertensive disorders are classified by the National High Blood Pressure Education Program Working Group on High Blood Pressure in Pregnancy into 4 categories: (1) chronic hypertension, (2) preeclampsia-eclampsia, (3) preeclampsia superimposed on chronic hypertension, and (4) gestational hypertension (transient hypertension of pregnancy or chronic hypertension identified in the latter half of pregnancy) [51].

Hypertension during pregnancy complicates 2–3% of all pregnancies. Gestational hypertension is defined as the new onset of hypertension after 20 weeks of gestation, without having hypertension before pregnancy, proteinuria, or other manifestation of preeclampsia/eclampsia. Preeclampsia develops hypertension after 20 weeks of gestation associated with proteinuria and is a multiorgan disease process [51]. The pathogenesis of preeclampsia is still studied and debated, but the most accepted is the immunological theory with abnormal placentation due to maternal intolerance to the semi self-fetus.

McDonnold and his team have conducted a retrospective cohort study in which have compared a group of pregnant women with HR-HPV infection (314) with a control group (628) and discovered a twofold higher risk of developing preeclampsia in the positive group (10.19% vs. 4.94%). However, the study has limitation and only reveal an association, not a causality, in order to prove causality, more extensive prospective studies are needed and adjustment secondary risk factors must be done: for ethnicity, smoking, BMI, chronic renal diseases or chronic high blood pressure [39]. Reily-Bell et al. [36] have recently published a work (2020) regarding Human Papillomavirus E6/E7 Expression in Preeclampsia and provided evidence that HPV E6/E7 is expressed in the placenta of women with preeclampsia When combining this finding with the findings that HPV infects trophoblastic cells since the first trimester, arises the potential role of HPV infection in the early placental development, being well known that preeclampsia has placental origins, but further investigations are needed to prove the determinant role on HPV in the malfunction of the placenta and therefore to prove that for sure HPV is a causative agent for pregnancy complications [45]. Contrariwise, Subramaniam in his retrospective cohort study, concluded that HPV is not an independent factor pregnancy-induced hypertension [38]. Due to the insufficient data linking maternal HPV infection with preeclampsia or other pregnancy-induced hypertensive disorders, HPV is not an established risk factor, and further investigations are needed.

#### 3.2.4. Fetal Growth Restriction (FGR)

Fetal Growth Restriction (FGR) is defined as the inability of the fetus to reach its biological growth potential, and it is diagnosed when the estimated ultrasonographic fetal weight is less than the 10th percentile for gestational age [52]. The pathogenesis for FGR is poor placentation. Based on additional biometrical measurements and the onset time FGR is classified into symmetrical and asymmetrical [52]. Symmetrical FGR represents 20–30% of all cases of FGR, is caused by genetic disorders or infections, and has an early onset before 32 weeks of gestation. Asymmetrical FGR begins after 32 weeks of gestation, constitutes 70–80% of all cases of FGR, has an increased ratio of HC/AC due to the “brain sparing” mechanism, and is caused by the uteroplacental deficiency [52].

It is notorious for specific infectious agents (TORCH) the causality with FGR. Going along with this parallel and having studies demonstrating the infection of trophoblastic cells with HPV [43,45] maternal HPV infection may be suspected to be a potential risk factor for growth restriction. Slatter and his coworkers conducted a cross-sectional study on 339 pregnancies and identified an HPV-associated lymphohistiocytic villitis manifested as patchy syncytiotrophoblast HPV positivity with lymphocytic and macrophage villous infiltration in addition to infection of the decidual tissue, and correlated this condition with a higher risk of FGR, prematurity, preeclampsia and fetal deaths similar to other infectious villitis with altering the inflammation pathway [43].

## 4. Discussions

There is a high interest in the correlation between HPV infection and pregnancy outcomes, and there are some studies that investigate this subject published at the date; most of them are investigating preterm birth, premature rupture of the membranes and the miscarriage rates.

Our systematic review is a clear reflection of the literature published in the last ten years on the involvement of HPV infection in adverse pregnancy outcomes and brings with it a brief presentation of all the conclusions from the included studies and a review of their limitations and potential bias Table 1.

Some infectious pathogens are implicated in pregnancy adverse outcomes by interfering with the intrauterine environment. The updated literature identifies multiple pathways that HPV may infect the conception product and its annexes—as by ascending infection from the cervix with contiguous infection of the uterine contents or as hematogenous disemmination [43,49]. Infecting the intrauterine environment, HPV creates an inflammatory reaction that can lead to a series of trigger reactions in the uterus, activating signaling pathways involving immune and hormonal biomarkers with a role in modulating uterine contractility and placental development. We further detail these pathways. Since HPV has been identified in the trophoblastic cells, decidual cells, placental cells, and amniotic fluid worldwide scientist are trying to understand the potential viral mechanisms that may alter the uterine homeostasis and generate adverse outcomes. Preterm birth occurs as a complex mechanism between the uterine myocytes and cell signaling by the steroid hormones secreted by the fetal hypothalamic–pituitary–adrenal axis. The trigger for this cell activation can be a mechanic factor (supradistention of the uterus—polihydramnios, macrosomia, multiple pregnancies, an incompetent cervix), an infectious factor (by generating inflammation), a genetic factor (family history of premature births), an imunological factor (low maternal tolerance of the semi-allogeneic fetus), an environmental factor, maternal stress. Intrauterine infections activate the innate immune system, release proinflammatory cytokines and chemotactic molecules with prostaglandins excretion, and stimulate uterine contractility. HPV, by its increased affinity for trophoblastic cells [43], in particular for the syncytiotrophoblastic cells which it infects in part [22], has a direct impact on pregnancy outcomes. HPV is decreasing both the number of trophoblastic cells affected by inducing apoptosis and the cellular adhesion to the maternal decidua leading to abnormal placentation and also affecting the quality of the embryo, and by these mechanisms increasing the risk for adverse pregnancy outcomes such as preterm birth; this risk is higher in HR-HPV infections [12]. This mechanism is especially proven for bacterial infections, but also viral infections seem to have similar mechanisms for triggering preterm birth. Maternal infections are also demonstrated as an etiological factor for abortion and premature rupture of membranes, and this theory can also be translated into maternal HPV infection. Certainly, the involvement of HPV infection in early pregnancy is not the only factor causing intrauterine damage, abnormal placentation, and increased risk for adverse pregnancy outcomes, and there is a need for further studies in cultures and model animals to identify hormonal and immunological metabolites with a potential biomarker of pregnancy risks as well as cofactors.

The pathophisiological pathway of preeclampsia and fetal growth restriction is based on abnormal placentation. As in vitro and in vivo studies have shown that HPV can infect the trophoblastic cells and therefore alter the early placental development and determine malfunction of the placenta, maternal HPV infection may be suspected as a potential risk factor for both preeclampsia and fetal growth restriction [25,30,36,40].

HPV may affect the pregnancy outcomes by affecting the cervical competence, altering the vaginal microbiota, ascending infection of the chorionic villi and causing placental dysfunction, infecting the amnitoic fluid causing amniotitis, or secondary to surgical treatments of the cervical HPV dysplasia; co-existence with other genital infection may also intefere with a normal pregnancy outcome; also semen infection may seem to have a potential role in generating adverse pregnancy outcomes. If there is a causative relationship between genital HPV infection and adverse pregnancy outcomes, we should see an improvement with the vaccination programs.

The studies included in our systematic review suggest that HR-HPV infection in pregnant women may be a risk factor for preterm rupture of membranes and preterm birth. Regarding miscarriage, fetal growth restriction, preeclampsia the small number of studies and the small sample sizes investigated prevents us from drawing any conclusions.Further studies are needed to complete the hypothesis that maternal HPV infection acts as a negative prognostic factor in pregnancy courses and to fully elucidate the pathophysiological mechanism of HPV infection in pregnancyIn addition to the preclinical studies needed to elucidate these mechanisms, retrospective observational studies are needed to identify potential cofactors of adverse pregnancy outcomes (ethnicity, BMI, tobacco, and alcohol use, hypertension, and chronic kidney disease in particular) [39].

## 5. Conclusions

HPV is the most prevalent infection in both women and men in their reproductive years, and it represents a global health problem when discussing genital cancers. This systematic review highlights the idea that maternal HPV infection during pregnancy may be a risk factor for adverse pregnancy outcomes like spontaneous miscarriage, prematurity, pregnancy-induce hypertensive disrders, fetal growth restriction, premature rupture of membranes, and even fetal death. Although there is a growing interest in this field and more and more studies are published there have been no clear conclusions yet and the data are still controversial. HPV infection is an infection that can impact both the mother and the fetus negativelyso further epidemiological studies are required to prove this causality. Also, standardized HPV testing protocols should be developed to have common conclusions when discussing the HPV causality of adverse pregnancy outcomes.

## Figures and Tables

**Figure 1 diagnostics-12-01471-f001:**
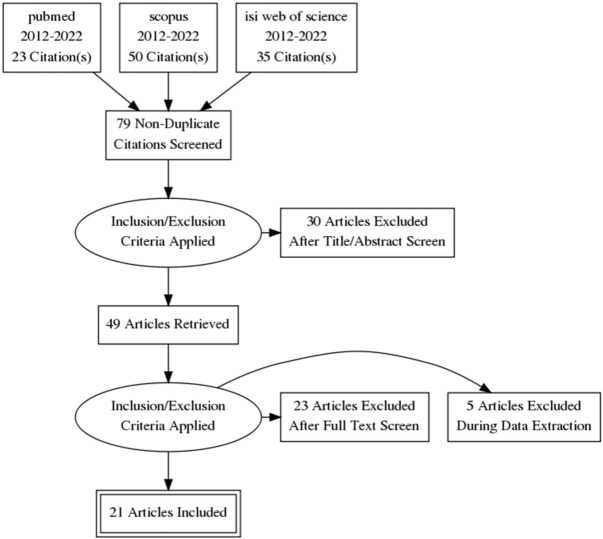
Flow diagram.

**Table 1 diagnostics-12-01471-t001:** Included studies—pregnancy outcomes and limitations.

Reference, Author, Year	Title	Journal	Study Design	No. of Patient	HPV Detection Tissue	Preterm Birth	Premature Rupture of Membranes (PROM)	Miscarriage	Pregnancy Induced Hypertensive Disease	Fetal Growth Restriction	Fetal Death	Study Limitation
Bober, 2019 [12]	Influence of human Papilloma Virus (hPV) infection on early pregnancy	Ginekol Pol	Case-control	143	Cervical, trophoblast, placenta			HR-HPV more prevalent among study group; *p* = 0.02				Limited sample size, limited statistical power
Wu, 2021 [13]	Systematic review and meta-analysis on influence of human papillomavirus infection during pregnancy on premature rupture of membranes and premature delivery	Ann Palliat Med	Systematic review and meta-analysis—7 studies included	45,603 22,799 = control group 22,799 = study group	Cervical, placenta	OR = 1.81, *p* < 0.05	OR =1.74, *p* < 0.05					Contradictory treatments and no randomization reports among the studies included, different retrieval mechanisms
Hornychova, 2018 [24]	Cervical human papillomavirus infection in women with preterm prelabor rupture of membranes	PLoS ONE	Case-control	100	Cervical, placenta		*p* = 1.00 HPV infection is not a risk factor					Limited sample size, no HPV detecting thru the amniotic fluid
Pandey, 2019 [23]	Human Papillomavirus (HPV) Infection in Early Pregnancy: Prevalence and Implications	Infect Dis Obstet Gynecol	Prospective study	104	cervical	*p* = 0.324	*p* = 0.026	*p* = 0.100	*p* = 0.470	*p* = 0.100		Limited sample size, only one time testing in the first trimester: new infection? And clearance during pregnancy?
Xiong, 2018 [29]	The Risk of Human Papillomavirus Infection for Spontaneous Abortion, Spontaneous Preterm Birth, and Pregnancy Rate of Assisted Reproductive Technologies: A Systematic Review and Meta-Analysis	Gynecol Obstet Invest	Systematic review and meta-analysis—18 studies included	6116	Cervical, placenta, amniotic fluid	HR-HPV OR: 2.84		indiscriminate genotype HPV infection OR: 2.24 meanwhile HR-HPV OR: 0.65				significant heterogeneity among the included studies; not report other adverse pregnancy outcomes risk factors
Niyibizi, 2020 [30]	Association Between Maternal Human Papillomavirus Infection and Adverse Pregnancy Outcomes: Systematic Review and Meta-Analysis	J Infect Dis	Systematic review and meta-analysis—36 studies included	342,796	Cervical, placental, amniotic fluid	aOR: 1.50	aOR: 1.96	aOR: 1.14	aOR: 1.24	aOR: 1.17	aOR: 2.23	significant heterogeneity among the included studies, no standardization among HPV testing, no clear identification of possible negative cofactors, misclassification of pregnancy outcomes
Ambühl, 2016 [31]	Human Papillomavirus Infection as a Possible Cause of Spontaneous Abortion and Spontaneous Preterm Delivery	Infect Dis Obstet Gynecol	Review 45 studies included	15,868	Cervical, placental, ombilical, amniotic fluid	*p* < 0.01		*p* < 0.05				significant heterogeneity among the included studies, inaccurate and inhomogeneous inclusion and exclusion criteria used, no cofactors investigated, different time of HPV testing
Chilaka, 2021 [32]	Human papillomavirus (HPV) in pregnancy—An update	Eur J Obstet Gynecol Reprod Biol	review			aOR: 1.5	aOR: 1.42	No association	No association	aOR: 1.17	aOR: 2.23	Heterogenic data, insufficient documentation of correlation between infection and adverse pregnancy outcomes
Condrat, 2021 [33]	Maternal HPV Infection: Effects on Pregnancy Outcome	Viruses	Systematic review 17 studies included	479,204	Cervical, placenta, amniotic fluid	x	x	x	x	x	x	Heterogenic data report, no statistical analyses only descriptive study
Basonidis, 2020 [34]	Human papilloma virus infection and miscarriage: is there an association?	Taiwanese Journal of Obstetrics and Gynecology	review	45,373				Unclear if there is any association				Descriptive study, no statistical analyses, heterogenic data, no cofactors investigated
Caballero, 2019 [35]	Maternal Human Papillomavirus and Preterm Premature Rupture of Membranes: A Retrospective Cohort Study	J Womens Health (Larchmt)	Retrospective Cohort Study	2153 829 HPV positive 1324 HPV negative	cervical	OR: 1.35, *p* = 0.04	OR: 2.07, *p* = 0.16		OR: 5.76, *p* < 0.001			interaction between HPV and other pathogenic organisms was not assessed in the study, limited population
Subramaniam, 2016 [38]	Evaluation of Human Papillomavirus as a Risk Factor for Preterm Birth or Pregnancy-Related Hypertension	Obstet Gynecol	retrospective cohort study	2321 242 HPV positive, 2079 HPV negative	cervical	OR: 1.3	OR: 1.7		OR: 1.0			Retrospective study, HPV testing identifying only HR-HPV, no data about HPV clearance, 3 years interval for HPV testing positive
Reily-Bell, 2020 [36]	Human Papillomavirus E6/E7 Expression in Preeclampsia-Affected Placentae	Pathogens	Case control	96	placenta				HR-HPV, *p* = 0.017; LR-HPV, *p* = 0.033			No other sexually transmitted disease detected into the placenta
McDonnold, 2014 [39]	High risk human papillomavirus at entry to prenatal care and risk of preeclampsia	Am J Obstet Gynecol	Retrospective cohort study	942	cervical	aOR:1.83, *p* = 0.04			HR-HPV aOR: 2.18, *p* = 0.004			Retrospective study, does not study causality, does not evaluate proteinuria or other co-factors involved in pathogenesis
Ticconi, 2013 [37]	Recurrent miscarriage and cervical human papillomavirus infection	Am J Reprod Immunol	Retrospective case-control study	524	cervical			Lower HPV infection prevalence in patients with recurrent miscarriage: 26.53% vs. 61.89%, *p* < 0.001				Retrospective study, different method of HPV detection
Ambühl, 2017 [45]	Human papillomavirus infects placental trophoblast and Hofbauer cells, but appears not to play a causal role in miscarriage and preterm labor	Acta Obstet Gynecol Scand	prospective case-control study	270	placenta	HPV prevalence in study group vs. control group:8.8%vs. 8.7%, *p* = 0.98		HPV prevalence in study group vs. control group: 10.9% vs. 20.4%, *p* = 0.19				Elective abortion as a control group
Mosbah, 2017 [40]	High-risk and low-risk human papilloma virus in association to spontaneous preterm labor: a case-control study in a tertiary center, Egypt	J Matern Fetal Neonatal Med	observational comparative case-control study	103	placenta	HPV prevalence in study group vs. control group 18.1% vs. 4%, *p* = 0.019						Limited sample size
Conde-Ferráez, 2013 [41]	Human papillomavirus infection and spontaneous abortion: a case-control study performed in Mexico	Eur J Obstet Gynecol Reprod Biol	Case control study	281	cervical			OR: 1.80, *p* = 0.0538				Limited sample size, no standardization regarding the moment of HPV detection
Cho, 2013 [42]	High-risk human papillomavirus infection is associated with premature rupture of membranes	BMC Pregnancy Childbirth	cross-sectional study	311	cervical	*p* = 0.718	OR: 2.380, *p* = 0.029		*p* = 0.054			Cross-sectional study, limited sample size, no cofactors investigated
Slatter, 2015 [43]	A clinicopathological study of episomal papillomavirus infection of the human placenta and pregnancy complications	Mod Pathol	Case control	339 253 HPV positive vs. 86 HPV negative	placenta	s.d 29.2% vs. 16.3%, OR: (odds ratio 2.13, *p* = 0.018			7.9% vs. 0%; OR: 8.4, *p* < 0.05	22.4% vs. 19.8%, *p* = 0.02	5.1% vs. 3.5%	no possible cofactors were identified, limited sample size heterogenous data
Aldhous, 2019 [44]	HPV infection and pre-term birth: a data-linkage study using Scottish Health Data	Wellcome Open Res	data-linkage study	5598	cervical	OR: 1.843, *p* = 0.020						No data about HPV treatment

## Data Availability

Data sharing is not applicable to this article as no new data were created in this study.

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
