# Peer review of "Maternal HPV Infection and the Estimated Risks for Adverse Pregnancy Outcomes—A Systematic Review"

_diagnostics, 2022, doi:10.3390/diagnostics12061471_

Round 1

Reviewer 1 Report

In the Systematic Review manuscript (diagnostics-1728333) entitled “Maternal HPV infection and the estimated risks for adverse pregnancy outcomes – a systematic review” by Dr. Popescu and colleagues investigated and analyzed the current literature on HPV infection and pregnancy, whit a focus for adverse pregnancy outcomes, such as preterm birth, miscarriage etc. A total of 21 eligible articles were analyzed. Main results indicate that although numerous studies on this topic being present, data are still controversial regarding identifying maternal HPV infection as a risk factor for adverse pregnancy outcomes

The meta-analysis respects the Preferred Reporting Items for Systematic Reviews and Meta-Analyses guidelines. The analysis is therefore well performed. Despite potentially interesting, the ms should be improved in terms of readability. For instance, the aim of the study should be included, while more supporting literature on the topic should be quoted. I have made some suggestions. Scientific writing style should also be improved. For instance, line 135 “observation studies” should be “ observational studies”. The citation style should be carefully checked and uniformed. A large variety of typo errors should be checked and corrected, as well. However, it should be noted that the manuscript will improve our general understanding on HPV infection and pregnancy in humans. In my opinion, the manuscript can be accepted following a major revision. Please find several suggestions for improving the ms.

MAJOR COMMENTS

1. Numbers identing the different sections, such as introduction, methods etc.., can be removed from the abstract

2. HPV footprints as DNA sequences alongside anti-HPV IgG antibodies have recently been identified in chorionic villi and sera, respectively, from both pregnant females and females affected by spontaneous abortion (https://doi.org/10.3390/vaccines8030473). As highly informative in terms of data (about 150 females were enrolled), this work should be at least included and/or mentioned in the manuscript. A detailed description, with additional studies, of the current status on HPV infection and pregnancy Is also described here DOI: 10.1093/humupd/dmv041

3. The causative role, if present, on HPV infection in preterm birth events, abortion as well as additional adverse pregnancy events, should be more deeply discussed in section 4. 

4. The aim of the work should be included at the end of the introduction

5. A large variety of typo errors should be checked and corrected.

6. The citation style should be carefully checked and uniformed.

7. Several sections are completely lacking in supporting references

MINOR

Line 34-36 A supporting reference should be included. For instance (doi: 10.3389/fonc.2019.00355)

Line 46 better starting the sentence with a word. For instance: “ A total of 40 subtypes…”

Lines 47-50. Infection were? The reported HPV-positie tissue should be mentioned 

Line 63, please include the period after the citation

Line 39 high risk and low risk HPVs are named HR-HPV and LR-HPV. Please revise the text accordingly

Line 46 HPV are also accountable for penile (https://doi.org/10.3389/fonc.2020.01521) and anal cancers DOI: 10.1002/ijc.2910430615

Line 67 please remove the double period before citations

Line 127 “in vitro” as well as other similar Latinism should be in italic. Please revise the entire text accordingly. 

Lines 128-129 early and late genes cannot be mentioned without a brief description

line 135 “observation studies” should be “ observational studies”

line 209 the space after “IVF” should be removed

Line 212-220 Please include supporting references

Lines 221-223 English should be improved

Line 270 The citation style should be carefully checked. 

Lines 339-356 This paragraph Is completely lacking in references. Supporting references should eb included

Author Response

Dear Reviewer,

Thank you a lot for your review! It was beneficial to us and helped us improve our work quality.
We have followed step by step all your recommendations from the major comments list as well as from the minor comments list.
We have checked and corrected the spelling and grammar errors.
We hope that our work meets your criteria and we will remain grateful to you!
Please see the attachment!
Your sincerely,
Andreea Boiangiu

Reviewer 2 Report

Type of manuscript: Systematic Review

Title: Maternal HPV infection and the estimated risks for adverse pregnancy outcomes – a systematic review

The systemic review provides comprehensive data about the estimation of the effect of HPV infection during pregnancy and assesses the correlation between HPV and adverse pregnancy outcomes.

1. Need to focus on the certain specific pathways /mechanism which is or are responsible for the adverse effect on pregnancy outcomes.

2. Do any other cofactors are involved in the pregnancy outcomes.

Author Response

Dear Reviewer,

Thank you a lot for your review! It was beneficial to us and helped us improve our work quality.
We have followed your recommendation from point 1 and we have detailed the specific pathways of HPV infection in pregnancy.
At point 2 you make an excellent observation about the cofactors involved in adverse pregnancy outcomes.
Unfortunately, the literature does not abound in articles that discuss these cofactors, as pointed out by a group of authors presented in our paper (Mac Donnold et al.).
For this reason we reiterate in discussions the need for such studies.
We hope that our work meets your criteria and we will remain grateful to you!
Please see the attachment!
Your sincerely,
Andreea Boiangiu

Round 2

Reviewer 1 Report

The manuscript can be accepted in the present form

Reviewer 2 Report

All the queries have been addressed. Would recommend it for publication.